# Enzymatic Pretreatment with Laccases from *Lentinus sajor-caju* Induces Structural Modification in Lignin and Enhances the Digestibility of Tropical Forage Grass (*Panicum maximum*) Grown under Future Climate Conditions

**DOI:** 10.3390/ijms22179445

**Published:** 2021-08-31

**Authors:** Emanuelle Neiverth de Freitas, Robson Carlos Alnoch, Alex Graça Contato, Karoline Maria V. Nogueira, Eduardo José Crevelin, Luiz Alberto Beraldo de Moraes, Roberto Nascimento Silva, Carlos Alberto Martínez, Maria de Lourdes T. M. Polizeli

**Affiliations:** 1Departamento de Bioquímica e Imunologia, Faculdade de Medicina de Ribeirão Preto, Universidade de São Paulo, Ribeirão Preto 14049-900, Brazil; emanuelleneiverthf@gmail.com (E.N.d.F.); alexgraca.contato@gmail.com (A.G.C.); karolmvnogueira@gmail.com (K.M.V.N.); rsilvausp@gmail.com (R.N.S.); 2Departamento de Biologia, Faculdade de Filosofia, Ciências e Letras de Ribeirão Preto, Universidade de São Paulo, Ribeirão Preto 14050-901, Brazil; robsonalnoch@usp.br (R.C.A.); carlosamh@ffclrp.usp.br (C.A.M.); 3Departamento de Química, Faculdade de Filosofia, Ciências e Letras de Ribeirão Preto, Universidade de São Paulo, Ribeirão Preto 14050-901, Brazil; ejcrevelin@ffclrp.usp.br (E.J.C.); luizmoraes@ffclrp.usp.br (L.A.B.d.M.)

**Keywords:** climate change, enzymatic pretreatment, laccase, dedicated energy crop, biorefinery, biofuels

## Abstract

Since laccase acts specifically in lignin, the major contributor to biomass recalcitrance, this biocatalyst represents an important alternative to the pretreatment of lignocellulosic biomass. Therefore, this study investigates the laccase pretreatment and climate change effects on the hydrolytic performance of *Panicum maximum.* Through a Trop-T-FACE system, *P. maximum* grew under current (Control (C)) and future climate conditions: elevated temperature (2 °C more than the ambient canopy temperature) combined with elevated atmospheric CO_2_ concentration(600 μmol mol^−1^), name as eT+eC. Pretreatment using a laccase-rich crude extract from *Lentinus sajor caju* was optimized through statistical strategies, resulting in an increase in the sugar yield of *P. maximum* biomass (up to 57%) comparing to non-treated biomass and enabling hydrolysis at higher solid loading, achieving up to 26 g L^−1^. These increments are related to lignin removal (up to 46%) and lignin hydrophilization catalyzed by laccase. Results from SEM, CLSM, FTIR, and GC-MS supported the laccase-catalyzed lignin removal. Moreover, laccase mitigates climate effects, and no significant differences in hydrolytic potential were found between C and eT+eC groups. This study shows that crude laccase pretreatment is a potential and sustainable method for biorefinery solutions and helped establish *P. maximum* as a promising energy crop.

## 1. Introduction

One of the most significant challenges humanity faces nowadays is building a sustainable future for a developing world. The increasing global consumption of fossil fuels by 50% in the next five decades, the price hikes, and the greenhouse gas emissions (i.e., CO_2_, CH_4_, N_2_O) generated by them, accelerate climate change and has shifted the research efforts on developing new alternatives of renewable fuels and chemicals [1]. The abundant availability and high holocellulose content of lignocellulosic biomass (LCB) provide a sustainable, cheap, and carbon-neutral emission option to produce renewable fuels and chemicals within the new biorefinery and circular economy concept [2].

However, the deconstruction of LCBs into fuels and chemicals has several drawbacks to their continued growth as a feedstock for energy production that must be overcome. Both microscale (e.g., lignin-carbohydrate complexes, cellulose crystallinity) and macroscale factors (e.g., food vs. fuel conflict, harvesting, and biomass consolidation) affect the production of bioproducts by lignocellulosic sources [3]. 

Lignocellulosic biomasses are mainly composed of carbohydrate polymers like cellulose and hemicellulose, which are hindered by lignin that is a heteropolymer composed of three main phenylpropane units: *p*-hydroxyphenyl (H), guaiacyl (G), and syringyl (S) [4]. Due to the close association of lignin with cellulose microfibrils forming lignin-carbohydrate complexes (LCCs), this polymer represents a significant barrier in the enzymatic conversion of polysaccharides into sugars that can be further fermented to produce second-generation ethanol or added value green chemicals. Furthermore, lignin is responsible for the unspecific binding of hydrolytic enzymes, decreasing the enzyme concentration during the saccharification process [5]. Therefore, a pretreatment step is often applied, and most of them aim to remove lignin in order to enhance carbohydrate’s accessibility and avoid enzyme losses [6].

Although there are several chemical and physical pretreatment methods for biomass delignification, all of them have limitations arising from the high energy input, saccharification and fermentation inhibitors, and the use of aggressive chemicals [7]. Thus, with the growing search for sustainable methods, the biologic pretreatment approach is a promising technology. It is an ecofriendly and cost-effective strategy due to its lower energy consumption and no release of toxic compounds to the environment. The biologic methods apply microorganisms and their biocatalysts that have an important role in the transformation of lignin due to their high substrate specificity contributing to process efficiency. In biological methods, the lignin degradation depends on lignolytic enzymes like lignin peroxidase (LiP, EC 1.11.1.14), manganese peroxidase (MnP, EC 1.11.1.13), and laccases (EC 1.10.3.2) [8].

Laccases are multicopper oxidoreductases that oxidize a broad range of phenols substrates with the benefit of using oxygen as the electron acceptor rather than H_2_O_2_. Laccases have been considered as a powerful tool in many biotechnological applications such as bioremediation processes, phenolics compound degradation, and for the enzymatic oxidation of lignin, offering advantages such as low-energy demand, milder pretreatment conditions, fewer side-reactions, lower fermentable carbohydrate loss during pretreatment, and environmentally safe process [9]. To date, laccases from white-rot basidiomycetes fungi have been described as the most efficient for biotechnological applications since they have the highest redox potential (0.8 V vs. Normal Hydrogen Electrode), which enable the enzyme to subtract electrons from the nonphenolic lignin subunits. An alternative to modulate redox potential is the use of low molecular weight compounds, known as mediators, whose enzymatic oxidation produces stable high redox potential intermediates that can further oxidize (nonenzymatic) other compounds that are not oxidizable by laccases alone [10].

Several studies have shown the efficient lignin removal, and increase in the reducing sugar content after saccharification in LCB pretreated with fungal laccases [11,12,13,14]. However, there are some challenges to using laccase for LCB delignification, such as high costs for enzymatic production, stability, and low-efficiency process, which makes the area of research still at an early stage in terms of commercial application [15]. Hence, more research is required to overcome the existing limitations in using these biocatalysts on an industrial scale.

In addition to developing new sustainable and economically feasible pretreatment methods, it is equally important to explore alternative lignocellulosic feedstocks to meet the global energy demand and overcome the drawbacks related to biomass harvesting, transport, and consolidation, which currently exists. For example, fast-growing perennial C4 grasses, such as Miscanthus spp. and switchgrass, have gained momentum as dedicated energy crops in the recent past, being widely used in Europe and the United States [16]. Likewise, *Panicum maximum* (Jacq. cv. Mombaça) is an important C4 grass used for pasture in Brazil, being rich in holocellulosic content (67%, *w*/*w*) and other several advantages that make it suitable for bioenergy purposes [17]. In addition, *P. maximum* has high efficiency in producing biomass through photosynthesis (30 ton/ha). It can grow in lands no longer utilized for agricultural uses, which minimize the food vs. fuel conflict and lead to cost-effective feedstock production [17,18].

Nonetheless, the climate change expected for the following years should affect growth, yield, and chemical composition since plant physiologic processes, such as photosynthesis, are sensitive to temperature and carbon dioxide (CO_2_) [19]_._ Thus, it is also necessary to evaluate the effects of climate change in C4 grasses as an energy crop after the enzymatic pretreatment step, once both are promising alternatives to the future of renewable fuels and chemicals.

We used a Trop-T-FACE system to simulate future temperature and CO_2_ atmospheric concentration under field conditions in the present work. Moreover, to achieve maximum fermentable sugar yield in the saccharification step, a response surface methodology was employed to optimize enzymatic pretreatment conditions using crude laccase from white-rot fungus. To study chemical reactions and physical rearrangements caused by pretreatment and further correlate fiber morphology, substrate composition, and lignin distribution with the enzymatic hydrolysis yield, structural (SEM/CLSM/FTIR), and compositional analysis were performed. We speculated that the crude laccase pretreatment processes could lead to chemical modification on holocellulosic composition and lignin depolymerization/modification, which might influence the climate changes effects on *P. maximum* and its potential as a dedicated energy crop. Thus, the present study comes up with approaches to address both macro and microscale factors that hamper the use of LCB for biorefinery purposes, also considering the climate change experts for the following decades, and thus, greatly helping in the establishment of lignocellulosic-based fuels and products.

## 2. Results

### 2.1. Evaluation Enzymes Produced by L. sajor caju in Orange Waste Solid-State Cultures

*Lentinus sajor caju* was cultivated in orange waste solid cultures for 14 days, at 28 °C. As expected, high levels of laccase activity (>20,000 U L^−1^) were obtained in the crude extracts (Appendix A). Manganese peroxidase activity was detected in small amounts (<180 U L^−1^), whiles lignin peroxidase activity was not detected in the crude extracts. Moreover, as manganese and lignin peroxidases required hydrogen peroxide (H_2_O_2_) for oxidation reactions, we assumed negligible activities for these enzymes in the crude extract. Appendix A also showed the activities detected for holocellulose complex enzymes in the crude extracts, and as expected, negligible activity values were also obtained for most enzymes evaluated. Thus, these results indicated the production of the laccase-rich crude extract by *L. sajor caju* under this culture conditions.

To confirm the production of laccases by *L. sajor caju,* two chromatography steps were used to purify laccases from the crude extract. As showed in Figure 1, a band with a molecular mass of approximately 52 kDa was evidenced in SDS-PAGE (Figure 1A). Zymogram analysis confirmed the laccase activity since only one active spot over agar-ABTS was observed (Figure 1B). The difference found in protein molecular weight for Zymogram is due to the non-denaturing conditions used, impacting relative protein mobility through the polyacrylamide gel. To confirm the identity of the enzyme, purified fractions corresponding to laccase activity were analyzed by mass spectrometry (LC/MS). The mass spectra data obtained were analyzed in the TPP server platform (Uniprot database) with the XTandem peptide search engine and ProteinProphet. The mass fragments analysis gave sequence coverages of 11% for the putative laccase 4 (Uniprot Q9HFT4) of *Lentinus (Pleurotus) sajor-caju* (Appendix A). Initial sequence analysis showed a theoretical molecular weight of 49.5 kDa, as evidenced in the SDS-PAGE (Figure 1A), confirming a rich-laccase crude extract produced by *L. sajor caju.* This extract, named LacLsc, was characterized and applied in the pretreatments used in the current work.

### 2.2. Laccase Activity Optimal Parameters

Both temperature and pH have great influence on enzymes activities. The pH affects active sites, which are composed of ionizable groups that must be in an acceptable ionic form to maintain the conformation of the active site and catalyze the reaction. Likewise, temperature affects the three-dimensional structure of proteins and their reaction velocity [20]. Therefore, we studied pH and thermal stability more closely to define the conditions to be applied on laccase pretreatment.

Figure 2A shows that pH 5.0 was the optimal value for *L. sajor-caju* laccase activity. Therefore the acetate buffer pH 5.0 was used for the pretreatment process. Regarding the thermal stability (Figure 2B,C), the crude laccase was stable in a temperature range of 40 to 50 °C, with a half-life time of approximately 1200 and 450 min for 40 and 50 °C, respectively. At temperatures above 50 °C, the activity drops off quickly in the first 30 min. In 24 h at 40 and 50 °C, the *L. sajor-caju* laccase presented a residual activity of 44 ± 1% and 23 ± 0.1%, respectively. Considering the above data, we kept pH 5.0 and pretreatment time at 6 h for the CCD experiment, while 41.6 to 58 °C was the temperature range tested in CCD experiments.

### 2.3. Optimization of Crude Laccase Pretreatment by Central Composite Design (CCD) and Response Surface Analysis

Laccase load (U g^−1^), mediator concentration (*w*/*v*), and temperature (°C) were investigated CCD since they are the main variables that influence the pretreatment efficiency and, therefore, the hydrolysis yield of fermentable sugars. The following reduced mathematical models (considering only the significant variables, *p* < 0.05) represent the influence of the independent variables on the final response, Sugar yield (g L^−1^). Equation (1) was obtained for *P. maximum* control group and Equation (2) for eT+eC treatment.
(1)$$Sugar\;yield\;\left(g\;L^{-1}\right)=10.98+1.29\;\left(X_{1}\right)-0.97\;{\left(X_{1}\right)}^{2}-0.91\;\left(X_{2}\right)-0.97{\left(X_{2}\right)}^{2}+0.73\;\left(X_{3}\right)-0.85\;{\left(X_{3}\right)}^{2}$$
(2)$$\begin{array}{c} Sugar\;yield\;\left(g\;L^{-1}\right)\\ =11.67+1.76\;\left(X_{1}\right)-0.69\;{\left(X_{1}\right)}^{2}-0.85\;\left(X_{2}\right)-1.04\;{\left(X_{2}\right)}^{2}+0.85\;\left(X_{3}\right)-1.47{\left(X_{3}\right)}^{2}\\ -0.47\;\left(X_{1}\right)\left(X_{3}\right) \end{array}$$
where *X_1_*, *X_2_*, and *X_3_*, correspond to the encoded values for laccase, mediator, and temperature, respectively.

The determination coefficient (R^2^) of the regression model was used to check its goodness of fit. The R^2^ for optimization of crude laccase pretreatment was 0.835 and 0.935 for C and eT+eC models, respectively. These numbers suggest the above models have just failed to explain 16.5% and 6.5% of the total variation. Although this number is more significant for the C group, it is considered a reasonable value considering biological samples. The ANOVA results of the second-order reduced models are summarized in Appendix A. The validation of the models arises when the variance (F) of the regression is greater than its tabulated F-value, as well the F value of the residues (lack of fit) is lower than its tabulated F-value. Thus, as both conditions were fulfilled, the models are valid for both climate conditions datasets (C and eT+eC).

The interaction between independent variables and their independent influence on sugar yields of *P. maximum* biomass after laccase pretreatment was graphically denoted by response surfaces. The contour curves (Figure 3) created based on the models described above show similar trends related to sugar yields (g L^−1^) for both climate conditions. The highest sugars yields found for C Figure 3 (A1–3) and eT+eC Figure 3 (B1–3) occur at intermediate to higher levels of laccase and temperature but in a lower concentration of Vanillin mediator.

The desirability profile (Appendix A) allows knowing the precise proportions of the independent variables that lead to the greatest levels of sugar release (response). According to this analysis for the C group, the optimum conditions for maximum sugar yield (g L^−1^) was achieved by the model at 228 (U g^−1^), 0.76 (% *w*/*v*), and 54.2 (°C) for laccase load, mediator concentration, and temperature (uncoded values), respectively. Likewise, to eT+eC group the optimum uncoded values were 350 (U g^−1^), 0.76 (% *w*/*v*) and 54.2 (°C). The laccase pretreated biomasses under these established optimum conditions were then submitted to hydrolysis assays to study its influence on the hydrolytic potential of *P. maximum*, grown under different climate conditions, as an energy crop.

### 2.4. Chemical Composition

Pretreatment methods are described as an important process to overcome LCB recalcitrance and increase the substrate accessibility to hydrolytic enzymes. In this study, enzymatic pretreatment was used to achieve these goals by targeting lignin removal and/or its structural modification. The chemical composition analysis of *P. maximum* grass in different climate conditions in native conditions (non-treated) and pretreated by crude laccase was carried out in terms of total anhydrousglucan, anhydrousxylan, anhydrousarabinose, anhydrousgalactose, lignin, and ash contents (wt %), as demonstrated (Table 1).

Analyzing the chemical composition (Table 1), two main effects were observed for pretreated substrates: (I) lignin removal and (II) cellulose-richer substrate, for both C, and eT+eC groups. The glucan content for laccase pretreated biomass was higher than the native material; the increase was from 26.2 to 30.4% glucan content for the C group, and 29.7 to 33.4% for the eT+eC group, on average. Moreover, laccase pretreatment was efficient for lignin removal achieving 40.8 and 46.2% delignification for C and eT+eC, respectively. Concerning the content of other components, no significant changes were found between non-treated and pretreated biomass, except for anhydrousxylose content that decreased from 17.3 to 15.7% for eT+eC climate condition. Considering the solid recovery and the chemical composition of *P. maximum* before and after pretreatment, only 7.5 to 9.2% (depending on the climate condition) of its glucan component was solubilized during crude laccase pretreatment, which is important since glucose is the most important fermentable sugar.

### 2.5. Validation of Crude Laccase Pretreatment at Determined Optimal Conditions

At the optimum operating conditions for crude laccase pretreatment obtained from CCD, the maximum theoretically sugar yield predicted by the model was 11.47 and 12.39 g L^−1^ (Equations (1) and (2)) for C and eT+eC groups, respectively. To further validate the model, we carried out experiments at optimum responses conditions predicted by CCD. As a result, the optimum experimental sugar yield was 11.54 and 12.46 g L^−1^, for C and eT+eC, respectively, with a standard error of 0.34 and 0.16 (Figure 4), which are close to the predicted response of the statistical model. Considering these values, a significant improvement of 41.2 and 36.6% in the sugar yields (g L^−1^) was found for laccase pretreated substrate compared to non-treated substrates (Figure 4A) at the same hydrolysis conditions (5% solid load and 15 mg protein/g biomass). The glucan conversion (%) was also performed to reveal the pretreatment process efficiency and it was calculated as a percentage of the theoretical glucan presented in the substrate and expressed as glucan conversion rates. For this experiment, the Ctec^®^ cocktail was applied in relation to the substrate glucan content (50 mg protein per gram of glucan). As a result, the glucan conversion for pretreated samples were 26.5 and 20.5% higher than non-treated substrates for C and eT+eC group, respectively (Figure 4B).

To further prove the positive effect of crude laccase pretreatment on hydrolysis sugar yields, we compared the sugar amount obtained for laccase pretreated with the ones resulted after pretreatments applying commercial laccase and inactive crude laccase (Figure 5). The experiments were performed at the same optimum parameters previously obtained in the CCD experiment and at the same hydrolysis conditions. As a result, the total sugar yields to inactivate crude laccase pretreated biomass, 7.81 and 8.48 g L^−1^ for C and eT+eC, respectively, were similar to the ones found in non-treated biomass, showing that the positive effects on sugar yields for laccase pretreated substrate are due to enzyme activity present on the crude extract of *L. sajor-caju*. Moreover, the commercial laccase pretreated samples showed significantly lower (*p* < 0.05) levels of sugar yields related to crude laccase, 9.8 and 10.26 g L^−1^ for C and eT+eC, respectively, which demonstrate that the use of crude laccase has costs and operational advantages.

Regarding climate conditions, significant differences between C and eT+eC groups were found just for non-treated and inactive laccase pretreated biomass. After commercial and crude laccase no significantly differences could be observed between these groups (Figure 5).

### 2.6. Hydrolysis Studies of Crude Laccase Pretreated P. maximum Biomass

#### 2.6.1. At Lower Protein Load

In addition to the barrier effect that lignin represents to the bioconversion of lignocellulosic feedstocks mainly due to the formation of lignin-carbohydrates complexes, this polymer has also been described to have another negative impact on enzymatic hydrolysis by promoting non-productive cellulase and xylanase adsorption through hydrophobic interactions [21]. As the latter effect is especially observed at low enzyme loadings [22], we carried out some hydrolysis experiments with reduced enzyme loading to ensure that 15 mg protein/g biomass had not “masked” any differences between the non-treated and pretreated substrates.

It was apparent that reduced protein load affects the differences found between non-treated and pretreated substrates (Figure 6). At 2 and 5 mg protein/g biomass, we found more significant relative increments on glucose and xylose releases after laccase pretreatment than non-treated biomass. At 2 mg/g biomass of protein load, the glucose and xylose releases were 41.0 and 50.1% higher for pretreated biomass of the C group and 40.5 and 57.1% for eT+eC treatment. Considering 5 mg protein/g biomass, the relative increment on glucose release for pretreated C and eT+eC were 39.1 and 35.4%, respectively, while xylose yield was 46.7 and 40.9% higher. For 10 mg protein/g biomass, the glucose and xylose yields were 29.2 and 34.4%, respectively, greater than non-treated C and 24.8 and 27.45% for eT+eC groups. Finally, at 15 mg g^−1^ of protein load, we observed similar differences to 10 mg g^−1^. Pretreated substrates exhibited 31.1 and 20.3% more glucose in the hydrolysate for C and eT+eC group, respectively, and 14.8 and 13.35% more xylose in relation to non-treated biomass. Notably, differences in glucose and xylose yields between non-treated and pretreated substrates were higher at hydrolysis with lower cellulase load, suggesting that laccase plays a role in reducing the hydrophobic interactions that result in non-productive biding on lignin that will be further discussed.

Nonetheless, the arabinose yield (g L^−1^) seems not to grow proportionally with the protein loading for both non-treated and pretreated substrates, and, therefore, its relative increment on laccase pretreated samples was almost constant at all conditions, being 45.3 and 27.3% (in average) higher for C and eT+eC pretreated biomass, respectively.

#### 2.6.2. At Different Solid Load

LCB hydrolysis must be performed at a commercial scale at higher solid loading to increase sugar concentrations and, therefore, ethanol/bioproducts yields. However, work with a higher solid load is challenging since rheological problems start to occur, and achieving proper mixing during the saccharification process turn to be a difficult process [23]. Thus, to study laccase contribution to overcome this challenge, we performed hydrolysis at higher solid loading. *P. maximum* in both climate conditions was pretreated for 6 h with crude laccase at optimal conditions, washed, dried, and hydrolyzed at different solid loads. The monosaccharides in the hydrolysate (glucose, xylose, and arabinose) were quantified through HPLC, and the results are presented in Figure 7.

The pretreatment with crude laccase had a significantly positive effect (*p* < 0.05) in relation with non-treated biomass for all tested solid load (2, 5, 8, and 10%) and for all monosaccharides analyzed (glucose, xylose, and arabinose) (Figure 7). However, we observed a minor relative increment at 2% solid load, in which the pretreated biomass showed 20.2 and 11.5% higher glucose release for C and eT+eC, respectively, compared to non-treated substrates. At higher solid load, the positive effect of laccase pretreatment increases; at 5% solid load, the relative increment in glucose, xylose, and arabinose were 36.7, 59.9, and 69.3%, respectively, for the C group. Likewise, for eT+eC, the relative increment in glucose, xylose, and arabinose was found to be 29.6, 31.6, and 42.4, respectively. At 8% solid load, the relative increment is even higher 37.54, 64.02, 80.3% for the C group, and 31.2, 31.21, and 57.66% for glucose, xylose, and arabinose yields, respectively, of eT+eC treatment. The highest relative increment was found at 10%, in which the C group showed 50.15, 90.2, and 97.7%, and eT+eC 32.67, 36.35, and 69.64% higher glucose, xylose, and arabinose release, respectively. These results demonstrate that the crude laccase pretreatment approach could reduce costs and increase the efficiency of the bioconversion process, especially at higher solid load; its effects are independent of the future expected climate changes.

The above-described increments for laccase pretreated biomass result in significantly higher (*p* < 0.05) concentration of fermentable sugars in the hydrolysate yielding from 3.6 to 20.5 g L^−1^ of glucose for the C group and from 3.7 to 20.7 g L^−1^ for the eT+eC group, depending on the solid load applied. In addition, xylose was released in satisfactory concentration for both C and eT+eC groups, reaching 3.4 and 3.5 g L^−1^, respectively, at 10% solid.

### 2.7. Simon’s Staning

Earlier works have used Simons’ staining method with the adsorption of direct orange (DO) dye to estimate the overall porosity/accessibility of pretreated lignocellulosic substrates since DO dye has molecular weight like cellulases and affinity for cellulose [22,24]. Thus, after the incubation with DO dye, it was possible to determine the amount of dye absorbed in the fiber, which is directly proportional to the enzyme accessibility. The results for Simon’s Staining for both non-treated and crude laccase pretreatment are presented in Table 2. The DO adsorption values for pretreated with crude laccase significantly increased (*p* < 0.05) for both climate conditions compared to the respective non-treated biomass, indicating enhanced overall cellulose accessibility after crude laccase pretreatment, from which the main effect is to modify and remove lignin. This result agrees with the hydrolysis data in which pretreated substrates showed the highest bioconversion rates (Figure 4, Figure 6 and Figure 7).

### 2.8. GC-MS Analysis of Lignin Degradation Products

After pretreatment of *P. maximum* with laccase-rich crude extract, 10 lignin degradation products were identified by GC-MS listed in Table 3 and their retention times. With the increase in pretreatment time, we observed the formation of new intermediates while some compounds found in initial reaction times were not detected later. Furthermore, as enzymatic reactions generate compounds according to enzyme-specific reaction mechanisms or substrate structural/chemical prosperities, some intermediates were found to be different between C and eT+eC climate conditions.

The intermediates found in this study have some lignin markers that include phenolic and aliphatic carboxylic acids, such as 2-ethylhexyl ester,3-phenylpropionic acid (No 2), 2-isopropyl-5-methylhexyl acetate (No 4), and 2-hydroxy-1,3-propanediyl ester-octadecanoic acid (No 10). These compounds are described to be derived from to phenylpropanoid class, which plays a role in linking lignin and carbohydrates [25]. Additionally, methylated derivatives of hydroxybenzaldehydes, as 2,4-dimethyl-benzaldehyde (No 1), could be released from the laccase catalyzed breakage of alkyl-aryl ether bonds in lignin and are associated sinapyl and coniferyl alcohol lignin precursors. In addition, the presence of the linear/branched oxygenated hydrocarbons (No 4, 6, and 10 in Table 3) suggests that lignin was further degraded by catalytic cleavage of C-O-C bonds.

Heterocyclic aromatic compounds such as 5,5,8a-trimethyl-3,5,6,7,8,8a-hexahydro-2H-chromene (No 7) and 5-methyl-indole (No 9), made of heteroatoms of oxygen and nitrogen, respectively, were observed. These intermediates could be a result of the radical polymerization process of phenylpropanoid units during laccase reactions. Another lignin degraded product in the form of phenolic ketone, 2,5-di-tert-butyl-p-quinone (No 8)was released during pretreatment, and it is usually produced by the oxidation of phenyl propane lignin monomers. The release of heterocyclic aromatic compounds and phenolic ketones positively affects on laccase pretreatment process since they are described as natural mediators that help non-phenolic lignin biodegradation [26], corroborating with the more than 40% delignification of *P. maximum* (Table 1).

The aliphatic compounds and linear alkanes found in this study (No 3, 4, 5, 6, and 10 from Table 3) resulted from ring-opening reactions through lignin depolymerization. These occurred when laccase-activated lignin subunits in the form of radicals are not stabilized by couplings reactions, but instead, they proceed to bond cleavage within lignin. This fact may lead to consecutive cleavages that facilitate ring-opening or lignin fragmentation [10]. Ring-opening products can further be involved in coupling reactions creating products such as the ones above mentioned.

An important finding from this study was that hydrolysis and fermentation inhibitors, as furfurals and hydroxymethyl furfurals produced in chemical pretreatment methods, were not observed in the GC-MS profile, illustrating that lignin has been specifically cleaved by laccase.

### 2.9. Physical Characterization of Optimized Laccase Pretreated Biomass

#### 2.9.1. CLSM and SEM Analysis

In order to understand the morphological and structural impacts of crude laccase pretreatment under optimum conditions on *P. maximum* biomass, we carried out scanning electron microscopy (SEM) and Confocal Laser Scanning Microscopy (CLSM) analyses. Unfortunately, the climate conditions were not approached for these techniques since it was not possible to note any differences in the images between these groups.

In SEM images, non-treated biomass (Figure 8A,B) is characterized by highly ordered and tightly packed fibers that are covered by a lignin layer at the surface. However, upon the crude laccase pretreatment, the fiber structure was strongly modified to less tightly, and ordered ones with the loss of the packed assembly and with the formation of pores in the cell wall surface (Figure 8C,D) showed. These effects are likely due to the partial removal of the lignin layer that held the fibers together, thus increasing the surface area of cellulose for cellulase accessibility and, therefore, the yield of fermentable sugars.

For a more detailed study about lignin in the cell wall before and after laccase pretreatment, we performed a CLSM analysis. Since lignin is a chromophore, its distribution can be imaged and analyzed by detecting its autofluorescence. Thus, CLSM images were obtained from the surface of non-treated and pretreated biomass (Figure 9). Is it possible to observed changes in the emission spectrum of lignin where non-treated biomass (Figure 9A,B) emits fluorescence in the blue and green range (450–570 nm) of the electromagnetic spectrum with a green blush emission color, while laccase pretreated samples (Figure 9B,C) shift the fluoresce emission toward longer wavelengths, mainly in the green range (500–570 nm). In addition, it was possible to observe a reduction in fluorescence intensity for pretreated samples with some fiber sites having almost no fluorescence detected. Spectral and intensity fluoresce changes between native and pretreated samples are likely related to lignin degradation and/or structure modifications in the cell wall, also noted in SEM images.

#### 2.9.2. FTIR Analysis

The changes in the functional groups for laccase pretreated biomass in relation to the non-treated substrate was qualitatively observed by FTIR analysis aiming to identify modification in lignin structure (Figure 10A,B). For the C group (Figure 10A), after laccase pretreatment, a notable decrease in the intensity peaks was observed at 665 cm^−1^ (aromatic C–H binding of lignin), 1629 cm^−1^ (C=O stretching vibration in conjugated carbonyl of lignin), and 3396 cm^−1^ (OH stretching of lignin). However, a slight increase was detected at 1250 cm^−1^ (C-O vibration related to G lignin) and at 2919 cm^−1^ (C–H stretching of lignin). Further, no significant change was found for some functional groups belonging to holocellulose components (898 and 1727 cm^−1^).

The FTIR spectrum for the eT+eC group is presented in Figure 10B. Significant decrease in the absorption peaks was detected at 665 cm^−1^ (aromatic C–H binding of lignin), 1385 cm^−1^ (phenolic hydroxyl groups), 1427 cm^−1^ (C–H in-plane deformation of lignin), 1514 cm^−1^ (C=C stretching vibrations of aromatic rings of lignin), and 1727 cm^−1^ (hemicellulose components). Further, no significant change was found for 898 cm^−1^ peaks (amorphous cellulose) and for 1250, 1630, 2929, and 3400 cm^−1^ lignin-related groups.

## 3. Discussion

The pretreatment step is essential to the valorization of lignocellulosic biomass. It is often applied to remove or modify lignin, increasing cellulose accessibility to biocatalysts that lead to the products of interest. Laccases are recognized for their activity against lignin using oxygen (O_2_) as a final electron acceptor [9], making them a promising alternative as a biocatalyst to be applied in the pretreatment process of LCB. Furthermore, white-rot fungi laccases are described to have high redox potential (around +730 mV to +800 mV), increasing their ability to act towards lignin compounds [10]. In this work, we used a laccase-rich crude extract produced from *L. sajor-caju* using orange waste as a carbon source. The use of crude laccase was an attempt to reduce the enzyme purification steps that add significant costs to the process.

The tropical forage grass *P. maximum* is a potential source of LCB. However, the climate changes expected for the next decades can significantly impact its chemical composition and structure. Thus, the laccase-rich crude extract was applied for the pretreatment of *P. maximum,* grown under expected future climate conditions, to evaluate the potential of pretreated laccase biomass as an energy crop for the next decades.

Central composite design (CCD) and response surface analysis are considered powerful and valuable strategies to optimize biological processes; it is also timesaving related to the traditional ‘one-factor-at-a-time,’ helping to reduce costs in the enzymatic process [27] effectively. Therefore, we applied this statistical design to optimize laccase pretreatment, studying laccase load (U g^−1^), mediator concentration (*w*/*v*%), and temperature (°C) as independent variables. The response (dependent variable) was the Sugar Yield (g L^−1^) (sum of glucose, xylose, and arabinose) released after the saccharification step of the laccase pretreated substrates.

We chose to work with sugar yields rather than delignification since lignin removal is not always correlated with the greatest fermentable sugar yield [28]. Lignin rearrangements, reallocation, and structural modification are also described to lead to higher saccharification yields [29,30], but these chemicals alterations would not be quantified as delignification percentages. Nonetheless, by quantifying the total sugar yields after hydrolysis of laccase pretreated biomass, it is possible to address all laccase effects that directly or indirectly impacts enzymes accessibility and end up increasing the sugar release from LCB. All factors that could influence sugar production, except those studied by the CCD, were kept constant in pretreatment and hydrolysis steps.

Laccase pretreatment at optimized conditions led to significant changes in *P. maximum* chemical composition. Significant lignin removal (greater than 40%) was observed due to laccase action on *P. maximum* biomass (Table 1). Laccase mechanisms for lignin depolymerization and degradation include: (I) Decrease the number of aliphatic side chains describe to be involved in β-O-4′ and β-5′ interunit linkages (most common bond in lignin), suggesting Cα–Cβ cleavage; (II) Significantly removal of the three lignin monomeric units, p-hydroxyphenyl (H), guaiacyl (G), and syringyl (S); (III) Oxidation in Cα of syringyl lignin, and (IV) Breaking down the p-coumarates and ferulates ester bonds with polysaccharides in lignin carbohydrate complexes [10,11]. Lignin degradation products found in GC-MS analysis (Table 3) have some lignin markers that corroborate with the laccase delignification mechanisms mentioned above.

The lignin removal obtained in this study is higher than those reported using commercial laccases or in laccase association with other pretreatment methods (Table 4). Al-Zuhair et al., (2015) [31] stated 9% of delignification with commercial *Trametes versicolor* laccase using HBT (hydroxybenzotriazole) as mediator. Rencoret et al., (2019) described 24% of lignin removal from *Paulownia fortune* after laccase-mediated pretreatment combined with alkaline extraction with NaOH [32], while Gutiérrez et al., (2012) reported up to 32% reduction in lignin content for Elephant grass applying the same combine pretreatment [33]. Crude laccase of white-rot fungi was also used for pretreatment resulting in 8% delignification of oil palm empty fruit bunch [34]. Rajeswari and Jacob (2020) [12] reported a 76% maximum delignification of aloe vera leaf rind. However, the amount of laccase applied was at least 3-folds higher than in this study.

In addition to the beneficial effects in chemical composition, the enzymatic pretreatment using *L. sajor-caju* crude laccase has several advantages. One of the main benefits of using a biocatalyst is the high carbohydrates retention in the solid fraction after pretreatment. In this work, we achieved efficient delignification with more than 73% solid and 90% cellulose recovery, likely because laccase acts specifically on lignin (Table 1), avoiding carbohydrates losses associated with physical-chemical pretreatments [35]. Therefore, enzymatic pretreatments may have overall process efficiency like those found in physical-chemical methods when considering the sugar losses in the process. Moreover, laccase pretreatment decreases the production of fermentation inhibitors, such as furfurals and organic acids, resulted from cellulose and hemicellulose degradation in physical-chemical methods [8].

The biochemical changes introduced by laccase on lignin minimize its physical and chemical barriers and maximize cellulase accessibility to cellulose fibers. As a result, we observed significantly greater total sugar yields and glucan conversation rates compared to non-treated biomass for both climate conditions, indicating greater efficiency of enzymatic hydrolysis. At 5% solid load, we obtained greater than 11 g L^−1^ sugar release and almost 60% glucan conversion for pretreated substrates (Figure 4), suggesting that crude laccase pretreatment is a potential alternative as a pretreatment strategy to enhance cellulose conversion into fermentable sugars. The saccharification performances described in this study are more outstanding than the ones reported to other laccase pretreated LCB like Elephant grass (5 g L^−1^) [33], sugarcane bagasse (44.6% glucan conversion) [11], and wheat straw (29% glucan conversion) [36].

Although cellulases and hemicelluloses production by *L. sajor caju* were observed to be negligible (Appendix A) in compassion to laccase, it appears that these enzymes together with natural mediators present in crude enzyme extract may have aided in the pretreatment process. Mediators of natural origin, like lignin degradation products, fungal secondary metabolites, and hemicellulose debranching enzymes could act synergistically with laccases to reduce biomass recalcitrance. Together with reducing purification costs, these synergistic effects represent the advantage of using crude enzymatic extracts rather than purified commercial enzymes. This is possibly why crude laccase pretreatment led to a greater increase in sugar release than the use of commercial laccase (Figure 5).

Delignification catalyzed by laccase increases sugar yields primarily by increasing the internal surface area, improving cellulase binding on cellulose. In addition, lignin is a moderate hydrophobic polymer due to its aromatic nature, resulting in the unproductive adsorption of cellulolytic enzymes through interactions with hydrophobic moieties in cellulases [21]. Therefore, laccase-catalyzed grafting is a term used to describe the attachment of low-molecular-weight compounds, mainly carboxylic acid residues, to the lignin surface through radical coupling that ends up in reducing hydrophobic lignin properties and, therefore, unproductive cellulase binding [10,37].

To evaluate the influence of laccase on lignin hydrophobicity, hydrolysis at limiting protein load was carried out. We observed that hydrolysis yields of laccase pretreated samples increased for all tested parameters in relation to non-treated biomass (Figure 6). However, the hydrophilization benefic effect was more pronounced at lower protein load (2 and 5 mg protein/g biomass), in which cellulase adsorption in lignin strongly impacts the amount of enzyme available to degrade polysaccharides. In this case, the differences in glucose and xylose yields between pretreated and non-treated samples were greater than at higher protein loading (10 and 15 mg protein/g biomass) since at the latter conditions; there were sufficiently amount of enzymes to catalyze cellulose and xylose conversion, reducing the negative effect of enzyme-lignin interaction. These results indicate that laccase-catalyzed hydrophilization effectively reduced the unproductive binding capacity of lignin of *P. maximum* substrate, which is particularly advantageous to hydrolysis at lower enzyme loadings.

The beneficial effect of laccase in sugar release due to delignification and lignin hydrophilization was also observed for high solid load hydrolysis. Cost-effective enzymatic bioconversion of LCB on a commercial scale requires high substrate concentrates. However, several factors such as limited mass transfer, end-products inhibition, and enzyme adsorption result in inefficient reaction mixing and poor enzyme distribution, reducing reaction rates. Crude laccase pretreatment seems to play an important role for overcame these challenges since up to 26 g L^−1^ total sugar yield was achieved at 10% solid load (Figure 7). Moreover, the differences between non-treated and pretreated substrates proportionally increased with increasing solid load. Therefore, we hypothesize that laccase-mediated delignification and modification in lignin chemical properties (e.g., oxidation, the addition of hydrophilic groups, and solubility) alters the rheological characteristics of *P. maximum* biomass with consequential lower viscosities, providing conditions properly to enzymatic hydrolysis while reducing unproductive enzyme adsorption.

Concerning the differences found between climate conditions, for non-treated *P. maximum* biomass Control (C) group had significantly (*p* < 0.05) lower sugar yields compared to elevated temperature + elevated CO_2_ (eT+eC) for all conditions performed in this study (Figure 4, Figure 5, Figure 6 and Figure 7). The possible explanations for these differences are described in detail in a previous study from our research group, in which similar hydrolysis results were found [38]. However, no differences were found between C and eT+eC groups after crude laccase pretreatment, suggesting that crude laccase pretreatment extensively modifies *P. maximum* fibers. Thus, the minor differences in chemical composition between C and eT+eC groups were not enough to result in significant differences in saccharification performance in the tested hydrolysis conditions.

Also, the relative increase in sugar yields between pretreated and non-treated biomass showed to be higher for the C group in relation to eT+eC substrates (Figure 4, Figure 5, Figure 6 and Figure 7). This result indicates that laccase action was more advantageous to the C group, probably because this climate condition has a more recalcitrant structure than eT+eC for raw biomass [38]. Therefore, laccase pretreatment assisted in overcome these differences approximating the bioenergetic potential of both climate conditions.

The finds of Simon’s staining, SEM, CLSM, and FTIR analysis support the greater hydrolysis yields for arabinan, xylan, and cellulose components in laccase pretreated biomass. The fibers microscopic changes (e.g., fibers disorganization and pores formation) in SEM images (Figure 8) have already been described for pretreatment methods. They are generally considered to result from lignin removal [8,26,30,39].

As lignin contains some endogenous fluorophores, particularly the monolignols, CLSM analysis can quickly identify possible alterations in fluorescence emission and intensity that suggest lignin removal and/or modification. CLSM images (Figure 9) showed changes in the emission spectrum between pretreated and untreated samples where the former had predominant emission in the green region. Previous studies reported that longer wavelength and, thus, lower energy emissions are correlated with the deconstruction of the well-structured lignin assembly in the plant cell wall [30,40].

In addition, the less lignin content observed in Table 1 is supported by the loss in fluorescence observed in laccase pretreated samples. In particular, the content of β-aryl ether linkages (β-O-4′), one of the linkages targeted by laccase, was noticed to positively correlate with fluorescence intensity [41]. Other factors, such as modification in lignin-carbohydrates complexes cross-linkages and alteration in monolignols linkages, are also described to impact lignin fluorescence. The reduction in fluorescence observed also corroborated with other studies that found similar results associated with lignin degradation after laccase pretreatment [42].

Our SEM and CLSM analysis agree with those found in FTIR (Figure 10) and CG-MS (Table 3) analysis, where the alterations in bands attributed to lignin suggest depolymerization and/or chemical modification. The major reason for the decrease absorption peaks of the FTIR spectrum has been reported to be caused by laccase cleavage of lignin side chains without many structural alterations of holocellulose components [8,12]. Moreover, we observed some differences in the FTIR spectrum between the climate conditions indicating that the laccase mode of action on lignin depends on the chemical characteristics of the substrate.

## 4. Materials and Methods

### 4.1. Reagents and Raw Materials

Orange waste used for laccase production was donated by a local restaurant and subsequently dried and milled. Cellic CTec2 enzymatic cocktail and commercial laccase (Novozymes NS-22127) were kindly donated by Novozymes^®^ (Bagsvaerd, Denmark). The chromatography columns Hiprep Q FF and Superdex 75 10/300 GL were acquired from GE Healthcare Life Science (Chicago, IL, USA). Aminex HPX-87P column and Precision Plus Protein TM Standards were purchased from Bio-Rad Laboratories (Hercules, CA, USA). Malt Extract Agar, Vanillin, 3,5-dinitrosalicylic acid (DNS), sodium carbonate, and the substrates for enzymatic activities, 3-ethylbenzothiazoline-6-sulphonic acid (ABTS), 2,6-dimethoxyphenol (DMP), xylan beechwood, carboxymethylcellulose (CMC), locust bean, debranched arabinan, β-glucan, p-nitrophenyl-α-L-arabinofuranoside, p-nitrophenyl-β-d-galactopyranoside, p-nitrophenyl-β-d-glycopyranoside, p-nitrophenyl-β-D-xylanopyranoside, and p-nitrophenyl-β-d-cellobioside, were purchased from Sigma-Aldrich (St. Louis, MO, USA). All reagents used for the assays were of analytical grade.

### 4.2. Growth of Panicum maximum under Simulate Future Climate Conditions

Trop-T-FACE was used to assess the effect of two variables involved in climate change expected in the upcoming decades (i.e., elevated CO_2_ atmospheric concentrations [CO_2_] and global average temperature increase). In the current work, tropical grassland *Panicum maximum* cv. Mombaça was used as a model of study. Initially, plants grown for 60 days were clipped at 30 cm above the ground and cultivated for 24 days under the following treatments: (I) ambient atmospheric [CO_2_] and temperature (annotated as Control: C) and (II) 600 ppm atmospheric [CO_2_] and +2 °C above ambient temperature (named here as elevated temperature + elevated CO_2_: eT+eC). Then, *P. maximum* leaves were milled at 20 mesh, dried at 40 °C for 24 h, and stored in a free humidity environment until analyses. Meteorological data from the whole growing season was reported in [43].

The miniFACE (free-air CO_2_ enrichment) system was used to increase the atmospheric [CO_2_] by 600 μmol mol^−1^; for this purpose, PVC rings of 2 m diameter punctured with micro holes fumigated the plots with CO_2_. The T-FACE (temperature free-air controlled enhancement) system was used to increase the canopy temperature to +2 °C more than the ambient canopy temperature. We used a randomized four-block design with which the experiment plot being warmed by six infrared heaters mounted on reflectors in a 2-m-diameter hexagonal pattern. The control system integrates the canopy temperature of C plots and then regulates the canopy temperature of eT+eC to 2 °C over the ambient canopy temperature in warmed plots [43,44]. The trop-T-FACE facility is located at the campus of the University of São Paulo, Ribeirão Preto, SP, Brazil. The levels applied in the treatment were chosen according to Intergovernmental Panel on Climate Change models [45].

### 4.3. Fungal Strain, Culture Conditions, and Enzyme Extraction

The white-rot fungi *Lentinus sajor caju* (Fr.) Singer- CCB 020 was obtained from the fungi collection of the Botanical Institute, SP, Brazil. It was maintained in a malt extract agar medium at 28 °C. The production of the crude extract rich in laccase was carried out through solid-state fermentation (SSF), according to Freitas et al., (2017) [46]. In brief, cultures of *L. sajor caju* were inoculated in 250-mL Erlenmeyer flasks containing 7 g of orange waste (berry and peel) and 80% initial humidity using mineral solution [47] and incubated for 14 days at 28 °C. Crude extracts were obtained by adding 20 mL of water to each flask. The mycelia were separated by gauze filtration. Finally, the supernatant was centrifuged at 9500× *g* at 4 °C for 10 min. The supernatants, named as the laccase-rich crude extract, were stored at −20 °C until use.

### 4.4. Enzyme Assays

Laccase activity was determined using 3-ethylbenzothiazoline-6-sulphonic acid (ABTS) as substrate. The ABTS oxidation was monitored by increasing absorbance at 420 nm (at pH 5.0 ε_420 nm_ = 36 mM^−1^cm^−1^). Under assay conditions, one unit of activity (U) was defined as the liberation of 1 μmol of product equivalents per min. In addition, ligninolytic enzyme activities were measured as previously described [46]. Cellulases and hemicellulases activities were also determined using their respective substrates (synthetic and naturals) (Appendix A), as describe by Contato et al., (2021) [2].

### 4.5. Protein Content Determination, Electrophoresis, and Zymogram Analysis

Protein content was determined using the ninhydrin assay [48], with bovine serum albumin as the standard. Electrophoresis analysis of protein samples was done with 12% SDS-PAGE carried according to Laemmli (1970) [49]. The gel was stained with Coomassie Brilliant Blue R-250. For the zymogram analysis, β-mercaptoethanol was not added to the sample buffer, and the samples were not heated before running. After electrophoresis, the gel was washed with a mixture (50:50) of isopropanol–acetate buffer (50 mmol L^−1^, pH 5.0) for 30 min, and once with acetate buffer for 30 min, to remove the SDS. The gel was transferred into a plate with an agar-ABTS layer (1% *w*/*v* agar and 0.05% *w*/*v* ABTS) and incubated at 25 °C until green bands appearance. The apparent molecular masses of proteins were calculated by comparing their electrophoretic mobility with standard protein markers (Precision Plus Protein TM Standards Bio-Rad).

### 4.6. Purification and Identification of Laccase by Mass Spectrometer Analyses

Two chromatography steps were used for laccase purification. First, the crude extract was loaded on an anion exchange column (Hiprep Q FF) equilibrated with 50 mmol L^−1^ phosphate buffer, pH 6.5, integrated into an ÄKTA Purifier 10 FPLC System UV−900 (GE Healthcare Life Science, Cranbury, NJ, USA) chromatography system. Proteins were eluted with an increasing gradient of NaCl (0.05 to 1 M), and the elution was monitored at 280 nm. Fractions of interest were selected by laccase activity and SDS-PAGE page, pooled, and concentrated in Vivaspin (GE Healthcare, Cranbury, NJ, USA) ultra-filtration devices with 10 kDa membrane. Next, the selected fraction was loaded on a size exclusion column (Superdex 75 10/300 GL) equilibrated with 50 mmol L^−1^ phosphate buffer, NaCl 150 mmol L^−1^, pH 6.5). Protein elution was monitored at 280 nm, and the fractions with protein of interest were selected as described above.

Characterization of purified samples was carried out using a Waters Xevo TQ-S mass spectrometer system. For analysis, 100 μg of purified protein samples were diluted in a solution of Tris-HCl 100 mmol L^−1^, CH_4_N_2_O 8.0 mol L^−1^, pH 8.5. Protein cysteine residues were reduced with 100 μg dithiothreitol (DTT) at 37 °C for 60 min and then alkylated with 300 μg iodoacetamide for 60 min. The protein solution was then diluted with Tris-HCl 200 mmol L^−1^ (pH 8.0) to reduce CH_4_N_2_O concentration before tryptic digestion (37 °C overnight).

Solid-phase extraction was performed in Oasis HLB Cartridges (Waters). The dried sample was dissolved in 50 μL of 5% acetonitrile solution. It was chromatographed three times in the Waters Acquity UPLC I-class with a 2–50% acetonitrile gradient in 60 min using 0.1% formic acid as a modifier. The mass spectra were acquired in the Waters Xevo TQ-S mass spectrometer in survey mode with masses for parent peaks ranging from 200 to 1800 and for child peaks from 200 to 1600 with a cone voltage of 50 V and collision energy of 20 eV. The files were converted to Mzxml format using Masswolf software, and the analysis was performed on the TPP server platform with the XTandem peptide search engine. Carbamidomethyl was inserted as fixed modification, and oxidation of methionine was inserted as variable modification. Protein identification was performed considering high probability values using ProteinProphet.

### 4.7. Effect of the Temperature and pH on Laccase Activity and Stability

To pre-establish, the optimum performance conditions for laccase activity in the biomass pretreatment, the effect of temperature and pH on laccase activity and stability was determined. The effect of temperature (40–80 °C) and pH (3.0–7.0) were determined by measuring the laccase activity using ABTS as substrate under standard conditions. Thermal stability assays were performed at pH 5.0. The residual activity was measured as the percentage of enzyme activity after a specific time in relation to the activity in the initial time (100%) was treated identically but without incubation.

### 4.8. Optimization of Laccase Pretreatment

Initially, *P. maximum* samples were washed six times with 80% ethanol and then with water until the soluble sugars were entirely removed to avoid soluble sugar interference during pretreatment hydrolysis experiments. All samples were then dried in an oven at 50 °C. For the pretreatment step, we used the laccase-rich crude extract produced by *L. sajor caju* as previously described.

The optimization of crude laccase pretreatment of *P. maximum* (Control and eT+eC treatment) was carried out using 2^3^ central composite design (CCD) and response surface methodology. The efficiency of laccase pretreatment was evaluated through the effect of laccase load (X_1_) (100–350 U g^−1^), mediator concentration (X_2_) (0–1.68% *w*/*v*), and temperature (41.6–58.4 °C) (X_3_) on the sugar yield (g L^−1^) as the dependent variable. The mediator used was Vanillin, and all pretreatment runs were conducted for 6 h at 2% solid loading (*w*/*v*) and pH 5.

The levels of the independent variables were defined based on a complete experimental design (data not showed). A total of 18 experiments were performed. Among them, 15 experiments were structured in a factorial design (including 8 factorial points, 6 axial points, and 1 central point) and 3 experiments represent the replicates of central points. Table 5 represents the uncoded and coded levels for each run in the CCD.

Response surface methodology was used to analyze the experimental data can be approximated by the quadratic polynomial as expressed below (Equation (3)).
(3)$$Y=\beta_{0}+\underset{j}{\sum }\beta_{j}x_{j}+\underset{i<j}{\sum }\beta_{\left\{ij\right\}}\;x_{i}x_{j}+\underset{j}{\sum }\beta_{jj}x_{j}^{2}$$
where: *Y* denotes the dependent variable (Sugar Yield), *β*_0_, *β_j_*, *β_jj_*, and *β_ij_* represent the constant co-efficient and *X_i_* and *X_j_* stand for the coded independent variables that have influenced the response of variable *Y*.

Following the pretreatments performed at the conditions established in Table 5 for X_1_, X_2,_ and X_3_, *P. maximum* pretreated biomasses were vacuum filtrated, washed with 0.1 L of water, and dried at 50 °C. Samples were then subjected to enzymatic hydrolysis using Cellic CTec2 enzymatic cocktail (Novozymes^®^, Bagsvaerd, Denmark) at 15 mg protein per gram of biomass. The hydrolysis was carried with an initial solid loading of 5% (*w*/*v*) of the laccase pretreated biomass in 50 mmol L^−1^ sodium acetate buffer (pH 4.8) at 50 °C for 48 h. The dependent variable (*Y*) was the sum of glucose, xylose, and arabinose obtained after the Ctec2 hydrolysis called here as Sugar Yield (g L^−1^). It is important to note that all paraments from pretreatment and hydrolysis were kept constants for all 18 runs except for the studied independent variables (laccase load, mediator concentration, and temperature).

### 4.9. Chemical Characterization

The chemical composition of *P. maximum* was determined for untreated, and laccase pretreated biomass in three independent replicates. The average composition analysis was established using the Klason lignin method derived from the TAPPI standard method T222 om-88 [50]. Ash content was measured according to the NREL/TP-510-42622 method [51].

Glucan and lignin recoveries in the laccase pretreated materials were obtained according to the following equation:(4)$$Recovery\;\left(\%\right)=\frac{DM_{PT}\;\times \;C_{PT}}{DM_{I}\;\times \;C_{I}}\times 100$$
where *DM_PT_* is the dry mass after pretreatment, *DM_I_* is the initial dry mass, *C_PT_* is the glucan or lignin content after pretreatment, and *C_I_* is the initial glucan or lignin content. Lignin removal was calculated by subtracting the lignin recovery from 100% (maximum yield).

### 4.10. Enzymatic Hydrolysis Studies of Crude Laccase Pretreated Biomass

To better understand the effects of crude laccase pretreatment on the potential of *P. maximum* as LCB, we performed enzymatic hydrolysis in different protein and solid loading. The experiments sets were carried out using raw (ethanol and water pre-washed) and crude laccase pretreated biomass at optimum conditions. For enzymatic hydrolysis studies, Cellic CTec2 was used (Novozymes^®^, Bagsvaerd, Denmark) in 50 mmol L^−1^ sodium acetate buffer (pH 4.8) for 48 h at 50 °C and 150 rpm. After hydrolysis, enzymes were centrifugated (10,000× *g* 5 min) and heated for 30 min at 100 °C for inactivation. The hydrolysate was stored at −20 °C for further analysis. We also performed hydrolysis with biomass pretreated by inactivated crude laccase (98 °C for 24 h) and commercial laccase (Novozymes NS-22127), all of them under optimized conditions.

### 4.11. Determination of Monosaccharides

The quantitative analysis of monosaccharides (xylose, arabinose, galactose, and mannose) in the Klason and saccharification hydrolysate was determined by a high-performance liquid chromatography system (HPLC- YL9100 model) equipped with an Aminex HPX-87P column at 85 °C, which was preceded with the appropriate pre-column and eluted with HPLC grade water at a flow rate of 0.6 mL min^−1^. The components were detected by a refractive index detector and quantified by external calibration.

### 4.12. Simon’s Stain

To measure the cellulose accessibility before and after laccase pretreatment, we applied Simon’s stain procedure using Direct Orange 15 (DO), according to previous studies by Chandra et al. [52]. For this experiment, 10 mg of raw and pretreated samples were mixed with PBS buffer and DO dye in increasing concentrations. The mixture was incubated at 70 °C and 180 rpm overnight. The tubes were then centrifuged, and the absorbance of supernatant at 450 nm was measured.

### 4.13. GC-MS Analysis of Lignin Degradatory Products

The pretreatment liquors were collected at different times and analyzed by GC-MS to analyze lignin degradation products generated by laccase activity. First, *P. maximum* biomass (C and eT+eC groups) were subjected to laccase pretreatment in the optimized conditions, and five hundred microliters of the supernatant were collected at 0, 1, 2, 4, and 6 h of pretreatment. The supernatants were then acidified to pH 1.0–2.0 by adding a few drops of 1 mol L^−1^ of HCl and 0.2 g of NaCl were then added. Next, the extraction was repeated three times using two volumes of ethyl acetate with vigorous shaking, the organic phase was separated, and 0.6 g of Na_2_SO_4_ was added. Finally, the volume of ethyl acetate was totally evaporated at ambient temperature, and the samples were resuspended in 100 µL of the same solvent. GC–MS analysis was carried out using a Shimadzu QP2010Plus mass spectrophotometer (Shimadzu Corporation, Kyoto, Japan) as described in detail in [53]. Lignin-degrading compounds were performed by comparing mass spectra to the NIST library (version 2.0) of the GC–MS.

### 4.14. Physical Characterization of Optimized Laccase Pretreated Biomass

#### 4.14.1. SEM and FTIR Analysis

Scanning electron microscopy (SEM) was conducted to analyze the surface morphology of *P. maximum* biomass before and after laccase pretreatment using a JSM-6610LV microscope. Dried samples were placed on an SEM stub and gold-coated before the observation. FTIR (Fourier transformed infrared spectroscopy) analysis in dried samples of raw and laccase pretreated biomass using FTIR spectrometer (Cary 600 Series, Agilent Technologies, Santa Clara, CA, USA). The samples were molded into KBr discs, and the FTIR spectrum was obtained by scanning within a range of 500–4000 cm^−1^. Lignin and carbohydrate peaks were chosen based on previous studies [8,26,39] and analyzed utilizing OriginLab version 8.0 software.

#### 4.14.2. Confocal Microscopy

Confocal laser scanning microscopy (CLSM) was applied to analyze lignin distribution with the cell wall before and after enzymatic pretreatment. CLSM allows mapping lignin by measuring its fluorescence emission spectra. The images from CLSM were collected from biomass surface using a Zeiss LSM 780 confocal microscope with a Coherent Chameleon laser (Tisapphire) for a two-photons (2 P) excitation and a Plan-Apochromat objective lens (20×) as described previously [5,30]. In addition, fluorescence measurements were performed for all non-treated and pretreated samples with the same acquisition parameters.

### 4.15. Statistics and Numerical Analysis

Data were expressed as the mean of three replicates ± standard deviations of the means (SD). The significance of the observed differences was tested using analysis of variance (ANOVA) from the Graph-Pad Prism 8.0 software (San Diego, CA, USA). The Tukey test availed differences among the studied parameters, and *p* values < 0.05 were considered statistically significant.

Statistica 8.0 was the software used for response surface methodology analysis. The response surfaces were generated once the statistical significance was determined by the analysis of variance (ANOVA) and using the F test (Fisher’s test).

To better understand the methods used, a flow chart of the experimental design is presented in Appendix A.

## 5. Conclusions

The steric hindrance imposed by lignin and its hydrophobic properties plays an important role in restricting cellulase access to cellulose and the unproductive binding of enzymes on lignin. Laccase pretreatment is a sustainable alternative to remove the lignin barrier to cellulose and has several advantages in relation to the conventional pretreatment process, such as specific activity towards lignin that reduces carbohydrate losses, besides the less production of fermentation inhibitors. Here, we applied a crude laccase of *L. sajor-caju* for the pretreatment of *P. maximum* biomass at the optimal conditions that led to significant lignin removal (up to 46%). The delignification and laccase-catalyzed lignin hydrophilization enabled *P. maximum* hydrolysis at up to 10% solid loading yielding up to 26 g L^−1^ of total fermentable sugars. In addition, for all tested conditions, enzymatic hydrolysis of arabinan, xylan, and glucan components greatly increased compared to non-treated biomass, especially at low cellulase loading and high solid load. Physical characterization of optimized laccase pretreated biomass through FTIR, SEM, CLSM, and CG-MS analysis agrees with hydrolysis results, showing clear evidence of lignin removal and chemical modification, showing fibers more accessible for enzymatic hydrolysis. Laccase pretreatment also contributes to mitigating the climate changes effects on the potential of *P. maximum* since no significant differences were found between C and eT+eC climate conditions after laccase pretreatment, showing that laccase could overcome higher recalcitrance properties found for non-treated C group due to modification induced on lignin and LCCs.

Therefore, our work contributed to the development of a technology that significantly reduces the operating costs of the pretreatment step since the use of crude laccase reduces protein purification costs, the method is operated under mild temperatures, and no chemicals are applied in the process. Furthermore, this study suggests that *P. maximum* is forage grass with a high potential for cellulosic ethanol production, showing similar or more excellent hydrolysis performance than many energy crops widely used worldwide.

## Figures and Tables

**Figure 1 ijms-22-09445-f001:**
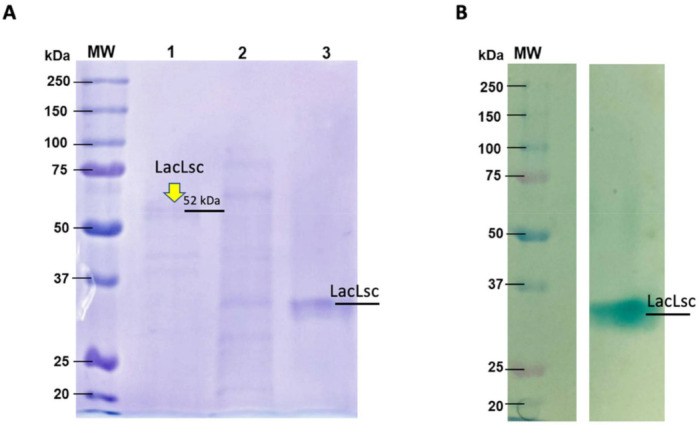
(**A**) SDS-PAGE of *L. sajor-caju* laccase purified through ion exchange followed by size exclusion chromatography (1), only ion exchange (2), and (3) native laccase (non-denatured) purified with ion exchange and size exclusion. (**B**) Zymogram in non-denaturing conditions visualized with ABTS-agar. The Bio-Rad standards contained 10 recombinant proteins (20–250 kDa). The SDS-PAGE gel was stained with Coomassie Blue R dye.

**Figure 2 ijms-22-09445-f002:**
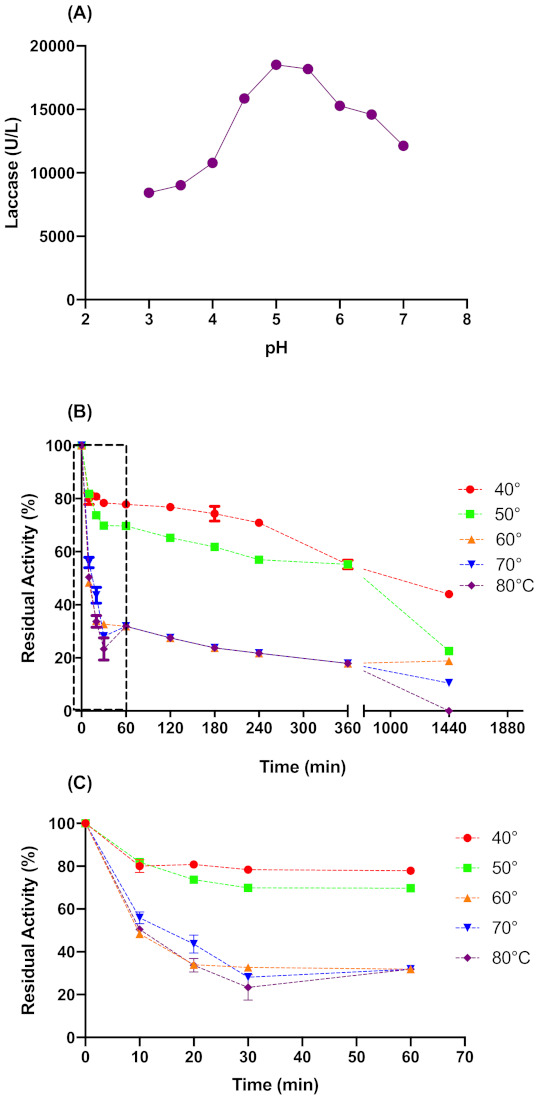
Optimum pH for catalytic activity (**A**) and thermal stability of *L. sajor-caju* crude laccase in longer times (**B**) and extension of shorter times (**C**). The buffer solutions for optimum pH assay were used according to their buffer range: Sodium Acetate (3–5) and Sodium Phosphate (6–8).

**Figure 3 ijms-22-09445-f003:**
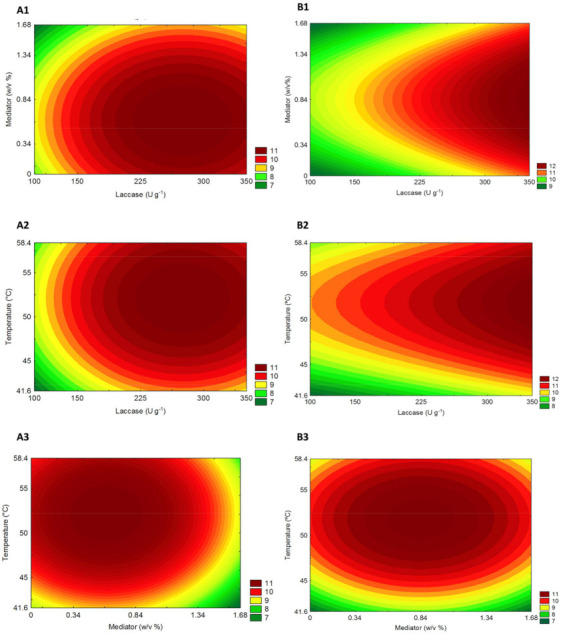
*Panicum maximum* response surfaces for the sugar yield obtained from hydrolysis after laccase pretreatment on (**A**) Control and (**B**) elevated temperature+elevated CO_2_ (eT+eC). Correlations are shown between (**1**) Laccase and mediator, (**2**) Laccase and Temperature, and (**3**) Temperature and mediator.

**Figure 4 ijms-22-09445-f004:**
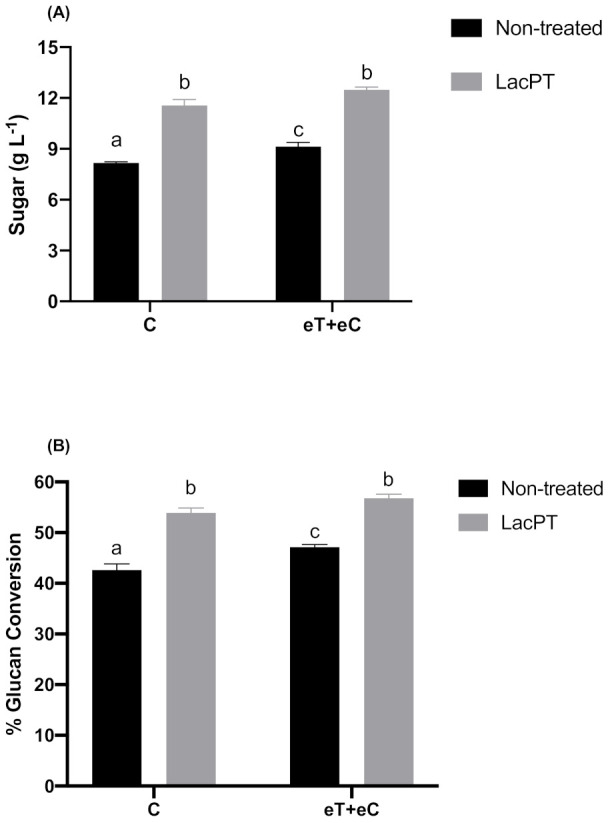
Sugar yield (g L^−1^) (**A**) and Glucan conversion (%) (**B**) after enzymatic hydrolysis (5% solid load and 48 h) for non-treated and pretreated *P. maximum* at different climate conditions. Treatments: Control (C) and elevated temperature + elevated CO_2_ (eT+eC). Mean values ± SD (*n* = 3) marked with different letters above bars symbolize statistical differences among treatments (*p* ≤ 0.05, Tukey’s test).

**Figure 5 ijms-22-09445-f005:**
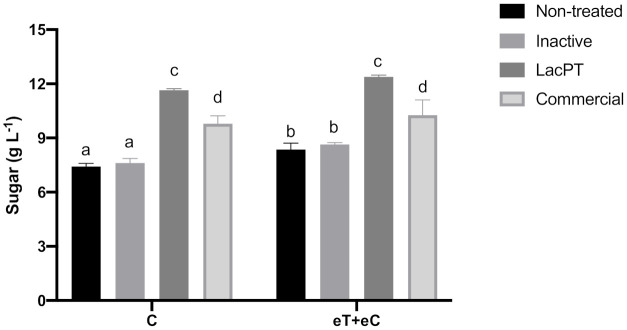
Sugar yield (gL^−1^) after enzymatic hydrolysis of non-treated and pretreated by inactive crude laccase, crude, and commercial laccase. Hydrolysis conditions: 5% solid loading, 15 mg protein/g biomass for 48 h. Treatments: Control (C) and elevated temperature + elevated CO_2_ (eT+eC). Mean values ± SD (*n* = 3) marked with different letters above bars symbolize statistical differences among treatments (*p* ≤ 0.05, Tukey’s test).

**Figure 6 ijms-22-09445-f006:**
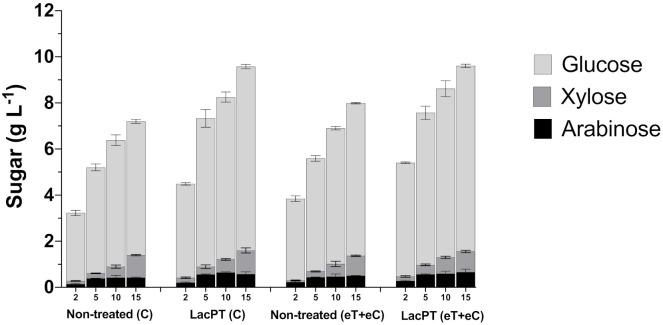
Glucose, xylose, and arabinose yield (g L^−1^) after enzymatic hydrolysis of non-treated (C and eT+eC) and laccase pretreated (C and eT+eC) biomass at 2, 5, 10, and 15 mg protein/g biomass. Hydrolysis conditions: 5% solid load for 48 h. Treatments: Control (C) and elevated temperature + elevated CO_2_ (eT+eC). Mean values ± SD (*n* = 3). The *p*-value was calculated using Tukey’s test and is discussed in the text at the proper place (*p*-value, significant < 0.05).

**Figure 7 ijms-22-09445-f007:**
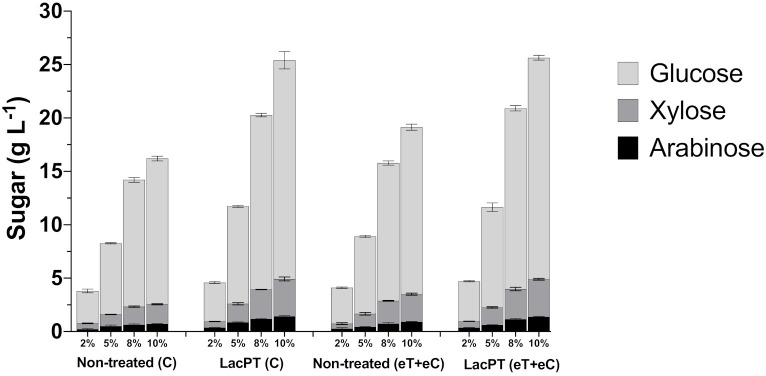
Glucose, xylose, and arabinose yield (g L^−1^) after enzymatic hydrolysis of non-treated (C and eT+eC) and laccase pretreated (C and eT+eC) biomass at 2, 5, 8, and 10% solid load. Hydrolysis conditions: 15 mg protein/ g biomass for 48 h. Treatments: Control (C) and elevated temperature+elevated CO_2_ (eT+eC). Mean values ± SD (*n* = 3). The *p*-value was calculated using Tukey’s test and is discussed in the text at the proper place (*p*-value, significant < 0.05).

**Figure 8 ijms-22-09445-f008:**
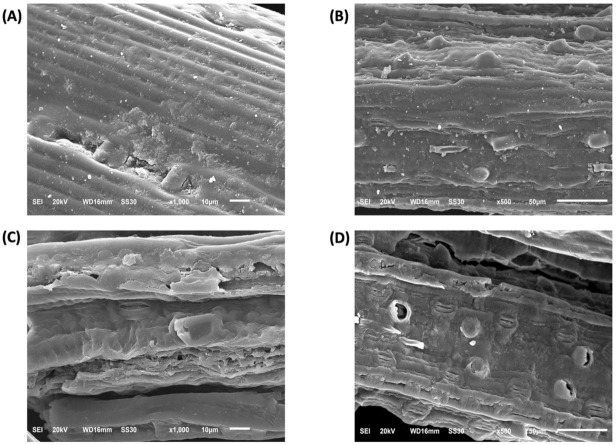
Scanning electron microscopy (SEM) images of (**A**,**B**) non-treated and (**C**,**D**) laccase pretreated *P. maximum* biomass.

**Figure 9 ijms-22-09445-f009:**
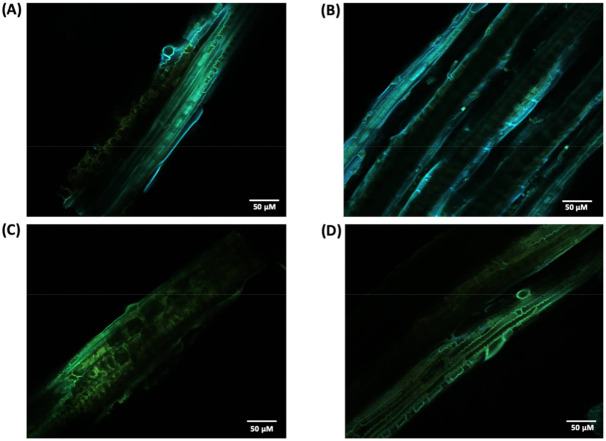
CLSM image of non-treated (**A**,**B**) and crude laccase pretreated (**C**,**D**) *P. maximum* biomass.

**Figure 10 ijms-22-09445-f010:**
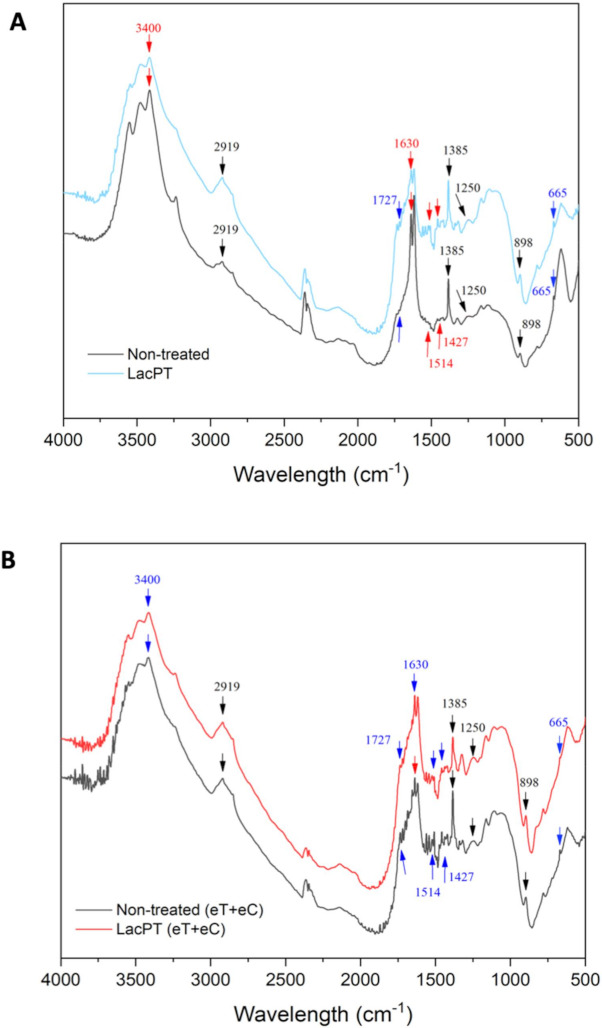
Fourier transform infrared spectra of raw and delignified pineapple leaf waste for Control (C) (**A**), and elevated temperature + elevated CO_2_ (eT+eC) (**B**). Pretreatment conditions: Laccase load of 228 U/g for C and 350 U/g for eT+eC, 0.84% (*w*/*v*) Vanillin and 54.2 °C for both groups.

**Table 1 ijms-22-09445-t001:** Percentage (%) of the chemical composition of non-treated and pretreated *P. maximum* samples in different clime conditions. Mean values ± SD (*n* = 3) marked with other letters at the same line are significantly different (*p* < 0.05, Tukey’s test).

Component (%)	Non-Treated		LacPT ^c^	
	C ^a^	eT+eC ^b^	C	eT+eC
Anhydrousglucose	26.2 ± 0.5 a	29.7 ± 0.1 b	30.4 ± 0.2 b	33.4 ± 0.3 c
Anhydrousxylose	17 ± 0.3 a	17.3 ± 0.2 a	16.3 ± 0.2 a	15.7 ± 0.4 b
Anhydrousarabinose	2.4 ± 0.12 a	2.3 ± 0.15 a	2.4 ± 0.12 a	2.2 ± 0.0 a
Anhydrousgalactose	1.15 ± 0.05 a	1.12 ± 0.03 a	1.08 ± 0.05 a	1.04 ± 0.07 a
Lignin	26.3 ± 0.8 a	29.8 ± 0.9 b	21.2 ± 1.2 c	20.5 ± 1.4 c
Ash	10.8 ± 0.2 a	11.5 ± 0.4 a	9.7 ± 0.4a	9.2 ± 0.6 a
Solid yield	N/A	N/A	73.4	78.2
Glucan recovery	N/A	N/A	92.5	90.8
Lignin removal	N/A	N/A	40.8	46.2

C ^a^: Control group. eT+eC ^b^: elevated temperature + elevated CO_2_. LacPT ^c^—Crude laccase pretreatment.

**Table 2 ijms-22-09445-t002:** Direct orange dye adsorption of non-treated and pretreated *P. maximum* samples in different clime conditions.

Biomass	Adsorption of Direct Orange (mg g^−1^)	
	C ^a^	eT+eC ^b^
**Non-treated**	63.7 ± 1.4 a	76.4 ± 1.7 a
**LacPT ^c^**	79.9 ± 3.0 b	84.7 ± 2.9 b

Mean values ± SD (*n* = 3) marked with different letters at the same column are significantly different (*p* < 0.05, Tukey’s test). C^a^: Control group. eT+eC^b^: elevated temperature + elevated CO_2_. LacPT^c^—Crude laccase pretreatment.

**Table 3 ijms-22-09445-t003:** Lignin degraded intermediates identified by GC-MS at different time courses of laccase pretreatment.

No	Intermediate Compounds	Control (C)			eT+eC			Retention Time (min)
		1 h	4 h	6 h	1 h	4 h	6 h	
1	2,4-imethyl-benzaldehyde	+	+	–	–	+	+	8.60
2	2-ethylhexyl ester,3-phenylpropionic acid	–	–	–	+	+	–	8.65
3	4,6-dimethyldodecane	–	–	–	–	–	+	8.81
4	2-isopropyl-5-methylhexyl acetate	–	–	–	–	–	+	9.59
5	2,3-dimethyldodecane	–	+	+	–	–	–	12.33
6	2-isopropyl-5-methyl-1-heptanol	–	+	+	–	–	–	13.39
7	5,5,8a-trimethyl-3,5,6,7,8,8a-hexahydro-2H-chromene	–	+	+	–	+	+	16.65
8	2,5-di-tert-butyl-*p*-quinone	–	–	+	–	+	+	17.84
9	5-methyl-indole	+	+	+	+	+	+	22.57
10	2-hydroxy-1,3-propanediyl ester-cctadecanoic acid	–	+	–	–	+	–	22.76

**Table 4 ijms-22-09445-t004:** Recent reports on several laccases capable of degrading lignin.

Laccase Source	Condition for Maximal Lignin Degradation	Delignification (%)	Reference
Commercial from *Trametes versicolor*	Use of HBT (hydroxybenzotriazole) as a mediator	9	[31]
Commercial from *Myceliophthora thermophila*	Laccase-mediated pretreatment with methyl syringate (MeS) combined with alkaline extraction with NaOH	24	[32]
Commercial *Trametes villosa* laccase	Laccase-mediated pretreatment (HBT) combined with alkaline extraction	32	[33]
Crude laccase from *Pycnoporus sanguineus*	Laccase-mediated pretreatment (HBT and ABTS) combined with alkaline extraction	up to 8	[34]
Crude white-rot fungi locally isolated	Laccase load (922 U g^−1^)	76	[12]
Crude laccase from *Lentinus sajor-caju*	Laccase-mediated pretreatment (Vanillin)	up to 46.2	This work

**Table 5 ijms-22-09445-t005:** Experimental design (conditions and responses) for crude laccase pretreatment of *Panicum maximum* biomass.

	Independent Variables			Sugar Yield for C Group (g L^−1^) ^d^		Sugar Yields for eT+eC Group (g L^−1^) ^e^	
Run Order	Laccase (X_1_) ^a^	Mediator (X_2_) ^b^	Temperature (X_3_) ^c^	Experimental	Predicted	Experimental	Predicted
1	150(−1)	0.34(−1)	45(−1)	8.46	9.03	8.34	8.95
2	300(+1)	0.34(−1)	45(−1)	9.94	10.33	11.2	11.18
3	150(−1)	1.34(+1)	45(−1)	8.3	8.11	8.21	8.56
4	300(+1)	1.34(+1)	45(−1)	9.13	9.41	9.88	9.85
5	150(−1)	0.34(−1)	55(+1)	9.91	9.76	9.54	9.8
6	300(+1)	0.34(−1)	55(+1)	10.77	11.06	11.56	12.03
7	150(−1)	1.34(+1)	55(+1)	8.33	8.84	9.42	9.41
8	300(+1)	1.34(+1)	55(+1)	9.48	10.14	10.77	10.71
9	100(−1.68)	0.84(0)	50 (0)	8.69	8.53	9.75	9.22
10	350(+1.68)	0.84(0)	50(0)	11.39	10.71	12.20	12.17
11	225(0)	0(−1.68)	50(0)	10.74	10.38	11.51	10.91
12	225(0)	1.68(+1.68)	50(0)	9.32	8.84	9.43	9.46
13	225(0)	0.84(0)	41.6(−1.68)	9.51	9.17	9.23	8.87
14	225(0)	0.84(0)	58.4(+1.68)	10.89	10.39	10.51	10.31
15	225(0)	0.84(0)	50(0)	11.03	10.98	11.84	11.67
16	225(0)	0.84(0)	50(0)	10.98	10.98	11.59	11.67
17	225(0)	0.84(0)	50(0)	10.52	10.98	11.40	11.67
18	225(0)	0.84(0)	50(0)	11.25	10.98	11.75	11.67

^a^ (U g^−1^), ^b^ (% *w*/*v*) and ^c^ (°C). C ^d^: Control group. eT+eC ^e^: elevated temperature + elevated CO_2_ treatment.

## Data Availability

Not applicable.

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
