# Peer review of "Enzymatic Pretreatment with Laccases from Lentinus sajor-caju Induces Structural Modification in Lignin and Enhances the Digestibility of Tropical Forage Grass (Panicum maximum) Grown under Future Climate Conditions"

_ijms, 2021, doi:10.3390/ijms22179445_

Round 1
Reviewer 1 Report
This is an outstanding study. I am amazed and admired by the authors' planning throughout this study. I suggest that this article can be accepted after minor revisions. Only a few suggestions are as follows:
- The so-called "future climate condition" needs to be clearly defined in the methods, instead of just writing "described in detail elsewhere [20,21]" and "according to Intergovernmental Panel on Climate Change models [22]".
- A concern of this study is about the future climate condition. Why do the authors set the conditions 600 ppm CO2 concentration and +2°C as a representative of future climate warming? The 600ppm CO2 concentration roughly matches the plateau phase of lower emissions (RCP4.5), which is a relatively mild warming scenario. Have you considered other warming scenarios?
- The contents of the experiments are large and a little cumbersome, and it isn't easy to understand the purpose of each experiment and their relationships before reading the discussion and conclusion. Giving a simple diagram of the research design (e.g., a flow chart of the experimental design) can facilitate readers better understand the necessity and value of each experiment in this entire research.
- L973: "od" should be "of."
Author Response
Reviewer 1:
Dear Dr, firstly, we value the time dispensed by the Editor and reviewers, and we thank all suggestions to improve our manuscript. We inform you that the text of our original manuscript was not changed in the revised version (normal black font), but the changes were made in red font. Our replies in this document are in blue font. We have also read carefully through the text and corrected any other minor mistakes that we have found (but without highlighting them). For better understanding, we list the comments by numbering them according to their sequence in the reviewers’ questions followed by their respective answers.
Dr. Maria de Lourdes Polizeli
This is an outstanding study. I am amazed and admired by the authors' planning throughout this study. I suggest that this article can be accepted after minor revisions. Only a few suggestions are as follows:
Comment 1: The so-called "future climate condition" needs to be clearly defined in the methods, instead of just writing "described in detail elsewhere [20,21]" and "according to Intergovernmental Panel on Climate Change models [22]".
R: We thank the reviewer for the valuable comment. Details of the TROP-T-FACE system and the levels chose were added to Section 2.2 in Materials and Methods. Please see lines 161-185.
Comment 2: A concern of this study is about the future climate condition. Why do the authors set the conditions 600 ppm CO2 concentration and +2°C as a representative of future climate warming? The 600ppm CO2 concentration roughly matches the plateau phase of lower emissions (RCP4.5), which is a relatively mild warming scenario. Have you considered other warming scenarios?
R: Thank you for your comment. According to IPCC RCP4.5 and RCP6.0 are two intermediate scenarios. Warming is likely to exceed 2°C for RCP6.0 and RCP8.5 (high confidence), more likely than not to exceed 2°C for RCP4.5 (medium confidence), but unlikely to exceed 2°C for RCP2.6 (medium confidence). Regarding the CO2 concentration, in RCP2.6 the CO2 concentrations will peak and decline and remain below 500 ppm, but as you mentioned the CO2 concentrations will reach 500 to 700 ppm in the RCP4.5 scenario. On the other side, CO2 concentrations will be above 700 ppm but below 1500 ppm, in RCP6.0 and RCP8.5 scenarios. In the study, we considered that warming is likely to exceed 2°C (RCP6.0 scenario) and the RCP4.5 scenario for increase in CO2 concentration.
Comment 3: The contents of the experiments are large and a little cumbersome, and it isn't easy to understand the purpose of each experiment and their relationships before reading the discussion and conclusion. Giving a simple diagram of the research design (e.g., a flow chart of the experimental design) can facilitate readers better understand the necessity and value of each experiment in this entire research.
R: We comply with the reviewer's request and added a flow chart of the experimental design in the Supplementary Material, as Figure S1, which was cited in the text in line 398.
Comment 4: L973: "od" should be "of."
R: We comply with the reviewer's request. Please see line 1037.

Reviewer 2 Report
Dear authors,
I have made my suggestions as pdf markings.

Author Response
Reviewer 2:
Dear Dr, firstly, we value the time dispensed by the Editor and reviewers, and we thank all suggestions to improve our manuscript. We inform that the text of our original manuscript was not changed in the revised version (normal black font), but the changes were made in red font. Our replies in this document are in blue font. We have also read carefully through the text and corrected any other minor mistakes that we have found (but without highlighting them). For better understanding, we list the comments by numbering them according to their sequence in the reviewers’ questions followed by their respective answers.
Dr. Maria de Lourdes Polizeli
Reviewer 2
Comment, pag. 1: .I suggest a better description of the system... although it may seem weird and redundant !!
R: Thank you for your comment. A better description of TROP-T-FACE system were added to material and methods section. Please see lines 161-185.
Comment, pag.2 : heavy chemicals could be replaced with aggressive chemicals or other words referring to substances such as strong acid and bases used for lignin removal.
R: Modified. Please see line 73 .
Comment: pag. 2: ....shown the efficient lignin removal..
R: Modified. Please see line 98.
Comment: pag. 2: of reducing sugar
R: Modified. Please see line 99.
Comment: pag. 3: for development of new
R: Modified. Please see line 106.
Comment: pag. 3: what do you mean by orange waste? I thought the paper was on P. maximum treatment...
R: We appreciate the comment. Orange waste was used for the culture medium of L. sajor-caju during solid state fermentation for laccase production (Line 145). P. maximum was the lignocellulosic biomass used in laccase pretreatment and hydrolysis experiments.
Comment: pag. 4: ... some explanation is needed on orange waste.
R: More explanation about orange waste was added to Material and Methods Section. Please see line 192.
Comment: pag. 4: what type of filtration system do you refer to...
R: We used gauze to separate the solid residues from the liquid extract containing extracellular enzymes such as laccase. Using hands, the gaze was pressed until maximum recovery of the liquid extract.
Comment: pag. 4: ..as described by ..."name of author" ..some more details might be of need ...I could not access reference [4]...
R: The name of the author was added, and the reference was checked (Line 206, actual ref 2. https://doi.org/10.3390/microorganisms9030533). However, we chose not to give more details about cellulase and hemicellulose activities since they are not the goal of this study and to simplify this part of methods.
Comment: this could be the subject of rearrangement to the beginning of a new page.
R: Modified.
Comment: pag. 5: characterization
R: Modified. Please see line 234.
Comment: pag. 5: GL standing for??
R: It was 80 % ethanol. We have changed GL to %. Please see line 264.
Comment: pag. 6: Response surface regression method ?? do you mean response surface methodology ? please clarify..
R: We mean response surface methodology. Modified in the text, please see line 284.
Comment: pag. 8: was established by using the following methods.
R: Modified in the text, please see line 313.
Comment: pag. 8: glucan only, not glucans...please specify if other polymeric sugars (xylan, arabinan etc) were determined and which was the method...
R: The method for determination of xylan, arabinan, and others is present below in Section 2.11 (Determination of monosaccharides), line 336.
Comment: pag. 8: ...do you mean in Klason supernantant? (hydrolysate)??
R: Yes, we mean Klason hydrolysate. We have changed it in the text, please see line 338.
Comment, pag. 9: Which software did you use... Statistica was mentioned earlier... but Graph Pad is mentioned here... in my opinion the sections on software used for data analysis could be combined.... if both were used.
R: Both were used, so as suggested we combined it in the Statistics and numerical analysis section, please see lines 394-399.
Comment, pag. 10: we have evaluated or better studied..
R: We have changed it in the text, please see line 441.
Comment, pag. 11: Residual activity could be defined
R: Residual activity definition was added to the materials and methods, please see lines 260-261.
Comment, pag. 11: which one is figure A, B and C... there is no annotation...
R: The A, B, and C denotations are above each graph, and also in the Figure 2 legend.
Comment, pag. 12: In the method description part you have used X1 for Lac, X2 for med, X3 for temp... some modifications are needed to respect the initial adnotations
R: Thanks for the comment. As suggested, the equations 3 and 4 were changed for X1, X2 and X3.
Comment, pag. 13: Color profiles could be adjusted to a better visibility of the levels.
R: The color profiles in Figure 3 were chose to follow the pattern applied for response surface representation in many studies. Below are some articles using the same color profile used in this study:
DOI:10.1016/j.ijbiomac.2021.07.065
DOI:10.1016/j.btre.2021.e00618
doi.org/10.1016/j.btre.2020.e00459
Comment, pag. 15: Do you mean anhydroglucose or glucan? same is valid for the rest of sugar components (xylan, galactan etc.)
R: We mean the monosaccharide, therefore, we changed glucan for glucose. Please see Table 2.
Comment, pag. 16: at determined optimal conditions
R: We have modified according to the suggestion. Please see line 557.
Comment, pag. 16: I think A and be could be combined in the same figure... as you did in figure 5.
R: Thanks for the valuable comment. Figure 4A and B are in different scales (Sugar g/L and Glucan Conversion %). Therefore, to combine the graphs, two Y axis would be needed. To make the figure clearly, we chose to keep it as A and B.
Comment, pag. 18: I suggest to combine the not treated with treated graphs for a better visibility of the differences...
R: R: As suggested we combined the non-treated with treated graphs for better visualization in both Figures 6 and 7.
Comment, pag. 30: The discussion on the effect of future climate conditions could be improved to better emphasize their impact...
R: The discussion on the effect of future climate conditions in conclusion was improved. Please see lines 1051-1053. More detailed discussion concerning to climate conditions is presented is lines 981-991.

Reviewer 3 Report
Manuscript: ijms-1356516
The manuscript by Freitas et al. “Enzymatic pretreatment with laccases from Lentinus sajor-caju induces structural modification in lignin and enhances the digestibility of tropical forage grass (Panicum maximum) grown under future climate conditions.” is interesting but lacks in novelty. This manuscript requires major revision to justify its significance.
Comments
- Lines 42-50, The authors should elaborate information about the quantum of fossil fuels, types, major greenhouse gases (such as CH4 and CO2) their emissions quantum, and biowastes such as lignocellulosic biomass availability i.e. DOI: 10.1016/j.rser.2021.111491.
- Line 55, biofuels? few examples.
- Lines 80-92, Please provide few examples of biotechnological applications of laccase such as phenolics compound degradation. “biotechnological applications since they have high redox potential” please specific the high, low, medium redox potentials of laccases and their benefits.
- Lines 57-63, The most complex component of lignocellulose is lignin, which is a hydrophobic heteropolymer composed of three major phenylpropane units: p‐hydroxyphenyl (H), guaiacyl (G), and syringyl (S) i.e. doi: 10.1002/biot.201800468.
- Lines 819-831, The authors should provide the key mechanism of lignin components degradation as a suitable illustration. Lignin degradation is widely reported in the literature by laccase; however, literature few comparatives may be provided to highlight the significance of the present study (as Table).
- Authors should provide the process economy analysis.
- At least one application i.e ethanol production additional data may be provided to justify the major claim.
- The quality of few images may be improved such as font size, line width, and resolution.
Author Response
Reviewer 3:
Dear Dr, firstly, we value the time dispensed by the Editor and reviewers, and we thank all suggestions to improve our manuscript. We inform that the text of our original manuscript was not changed in the revised version (normal black font), but the changes were made in red font. Our replies in this document are in blue font. We have also read carefully through the text and corrected any other minor mistakes that we have found (but without highlighting them). For better understanding, we list the comments by numbering them according to their sequence in the reviewers’ questions followed by their respective answers.
Dr. Maria de Lourdes Polizeli
Reviewer 3
Comment 1: Lines 42-50, The authors should elaborate information about the quantum of fossil fuels, types, major greenhouse gases (such as CH4 and CO2) their emissions quantum, and biowastes such as lignocellulosic biomass availability i.e. DOI: 10.1016/j.rser.2021.111491.
R: Thank you for the suggestion. We have elaborated the information and added the cited reference in the introduction. Please see lines 43-47.
Comment 2: Line 55, biofuels? few examples.
R: Thanks for the comment. We changed the word biofuels to bioproducts aiming at a broad approach to products that can be produced from lignocellulosic biomass. Please see the line 56.
Comment 3: Lines 80-92, Please provide few examples of biotechnological applications of laccase such as phenolics compound degradation. “biotechnological applications since they have high redox potential” please specific the high, low, medium redox potentials of laccases and their benefits.
R:. We provided some examples of biotechnological applications of laccase, and also added information about redox potential (Lines 90-91). The benefits of the high redox potential from laccase of white rot fungi are discussed in lines 92-97.
Comment 4: Lines 57-63, The most complex component of lignocellulose is lignin, which is a hydrophobic heteropolymer composed of three major phenylpropane units: p‐hydroxyphenyl (H), guaiacyl (G), and syringyl (S) i.e. doi: 10.1002/biot.201800468.
R: Information about lignin composition was added to the text and the suggested reference was cited. Please check lines 59-61.
Comment 5: Lines 819-831, The authors should provide the key mechanism of lignin components degradation as a suitable illustration. Lignin degradation is widely reported in the literature by laccase; however, literature few comparatives may be provided to highlight the significance of the present study (as Table).
R: We thanks the reviewer for the valuable comment. However, through the methods used in this study it is difficult to propose a mechanism for laccase catalyzed lignin depolymerization, further studies should be performed to clearly understand how L. sajor-caju laccase acts on lignin polymer. Nonetheless, based on literature, we proposed some possible mechanisms of laccase effect on lignin depolymerization and degradation, which is discussed in lines 865-877. We comply the reviewer request and add the Table 5, to compare the our results with other relevant literatures of laccase pretreatments.
Comment 6: Authors should provide the process economy analysis.
R: The reviewer raises a good point. However, economy analysis was not the purpose of this work, extra information about the large-scale application of this method is needed to fulfill this request. We have made in discussion and conclusion sections some considerations about the process economic viability. Please see lines 908-919, 932-942, 1054-1060.
Comment 7: At least one application i.e ethanol production additional data may be provided to justify the major claim.
R: Thanks for the comment. However, the purpose of this study was to optimize an enzymatic pretreatment by measure its effect on the hydrolysis yields. The sugar produced in the saccharification of pretreated biomass can be converted in a broad range of bioproducts through fermentation (i.e., chemicals, ethanol, organic acids). Therefore, we consider that we achieved the goal of this study by measuring the sugar release and compared the non-treated with treated groups along with the climate’s conditions.
Comment 8: The quality of few images may be improved such as font size, line width, and resolution.
R: The quality of images was improved for better visualization.

Round 2
Reviewer 3 Report
Accept as is